

# Immunofluorescence characterization of spinal cord dorsal horn microglia and astrocytes in horses

Constanza Stefania Meneses[1,*], Heine Yacob Müller[1,*], Daniel Eduardo Herzberg[1,*], Benjamín Uberti[2,*], Hedie Almagro Bustamante[2,*] and Marianne Patricia Werner[3,*]

[1] Veterinary Sciences Graduate School, Universidad Austral de Chile, Valdivia, Chile
[2] Veterinary Clinical Sciences Department, Universidad Austral de Chile, Valdivia, Chile
[3] Animal Science Department, Universidad Austral de Chile, Valdivia, Chile
[*] These authors contributed equally to this work.

## ABSTRACT

The role of glial cells in pain modulation has recently gathered attention. The objective of this study was to determine healthy spinal microglia and astrocyte morphology and disposition in equine spinal cord dorsal horns using Iba-1 and GFAP/Cx-43 immunofluorescence labeling, respectively. Five adult horses without visible wounds or gait alterations were selected. Spinal cord segments were obtained post-mortem for immunohistochemical and immunocolocalization assays. Immunodetection of spinal cord dorsal horn astrocytes was done using a polyclonal goat antibody raised against Glial Fibrillary Acidic Protein (GFAP) and a polyclonal rabbit antibody against Connexin 43 (Cx-43). For immunodetection of spinal cord dorsal horn microglia, a polyclonal rabbit antibody against a synthetic peptide corresponding to the C-terminus of ionized calcium-binding adaptor molecule 1 (Iba-1) was used. Epifluorescence and confocal images were obtained for the morphological and organizational analysis. Evaluation of shape, area, cell diameter, cell process length and thickness was performed on dorsal horn microglia and astrocyte. Morphologically, an amoeboid spherical shape with a mean cell area of $92.4 + 34\ \mu m2$ (in lamina I, II and III) was found in horse microglial cells, located primarily in laminae I, II and III. Astrocyte primary stem branches (and cellular bodies to a much lesser extent) are mainly detected using GFAP. Thus, double GFAP/Cx-43 immunolabeling was needed in order to accurately characterize the morphology, dimension and cell density of astrocytes in horses. Horse and rodent astrocytes seem to have similar dimensions and localization. Horse astrocyte cells have an average diameter of $56 + 14\ \mu m$, with a main process length of $28 + 8\ \mu m$, and thickness of $1.4 + 0.3\ \mu m$, mainly situated in laminae I, II and III. Additionally, a close association between end-point astrocyte processes and microglial cell bodies was found. These results are the first characterization of cell morphology and organizational aspects of horse spinal glia. Iba-1 and GFAP/Cx-43 can successfully immune-label microglia and astrocytes respectively in horse spinal cords, and thus reveal cell morphology and corresponding distribution within the dorsal horn laminae of healthy horses. The conventional hyper-ramified shape that is normally visible in resting microglial cells was not found in horses. Instead, horse microglial cells had an amoeboid spherical shape. Horse protoplasmic astroglia is significantly smaller and structurally less complex than human astrocytes, with fewer main GFAP processes.

Corresponding authors
Hedie Almagro Bustamante,
hbustamante@uach.cl
Marianne Patricia Werner,
marianne.werner@uach.cl

Instead, horse astrocytes tend to be similar to those found in rodent's model, with small somas and large cell processes. Microglia and astrocytes were found in the more superficial regions of the dorsal horn, similarly to that previously observed in humans and rodents. Further studies are needed to demonstrate the molecular mechanisms involved in the neuron-glia interaction in horses.

## INTRODUCTION

Glial cells have historically been considered as supporting cells for neuronal synapsis in the central nervous system (CNS), providing neurochemical precursors and energy sources, in addition to regulating the extracellular ionic concentration and removing cellular debris, among others (*Haydon, 2001*; *Watkins et al., 2007*). Interestingly, over the past two decades, the role of glial cells in pain modulation has gathered much attention (*Gosselin et al., 2010*). Scientific evidence has implicated microglia and astrocytes as key cells underlying acute pain and development of chronic pain after peripheral and central injury (*Hains & Waxman, 2006*; *Chen et al., 2012*). Painful syndromes, like traumatic spinal cord injury (SCI), are associated with glial response that can dynamically modulate the function of CNS neurons, increasing their excitability and reactivity (*Temburni & Jacob, 2001*). This differential functionality of glial cells is determined by the activation state of the cell and the components it expresses, which are different in duration of action and intensity according to the stimulus that triggers the glial reaction (*Watkins, Milligan & Maier, 2005*). Glial-derived mediators powerfully modulate excitatory and inhibitory synaptic transmission at presynaptic, postsynaptic, and extrasynaptic sites (*Ji, Berta & Nedergaard, 2013*), and moreover, are involved in the central modifications underlying chronic pain (*Gosselin et al., 2010*).

Glial activation is characterized by the upregulation of specific glial markers and/or morphological changes, including hypertrophy, proliferation, and modification of glial networks (*Ji, Berta & Nedergaard, 2013*). After nerve damage, active microglia display morphological changes, such as switching from ramified to amoeboid shapes (*Eriksson et al., 1993*) and may show upregulation of the ionized calcium-binding adaptor molecule-1 (Iba-1), a small calcium binding protein specifically expressed by these cells (*Hanisch & Kettenmann, 2007*). Several studies have demonstrated that morphological changes associated to Iba-1 expression in correlate with cell activation (*Ito et al., 1998*; *Faustino et al., 2011*). Furthermore, its overexpression is highly correlated with increased nociceptive inputs converging to the dorsal spinal cord days after injury (*Yamamoto et al., 2015*). Microglial activation seems to frequently occur during the early phase of peripheral or central lesions and may even precede astrocyte changes (*Tanga, Raghavendra & DeLeo, 2004*; *Hald et al., 2009*). Once activated, these cells can release a variety of neuroactive substances, including proinflammatory cytokines (TNF, IL-1, IL-6), nitric oxide (NO),

reactive oxygen species (ROS), arachidonic acid products, excitatory amino acids and ATP, increasing the excitability and reactivity of nociceptor neurons (*Thameem, Kaur & Ling, 2007*; *Faustino et al., 2011*; *Yamamoto et al., 2015*).

Similarly, astrocyte resting and active states have been defined. It is classically considered that resting astrocytes perform a constant surveillance function, and express basal levels of GFAP (*Gosselin et al., 2010*). Nevertheless, it appears that initial microglial activation after injury can lead to astrocyte activation (*Tanga, Raghavendra & DeLeo, 2004*). Activation of astrocytes generate morphological changes at the dorsal horn level, such as cellular hypertrophy and increased GFAP expression, leading to post-translational modifications that can increase the secretion of pro-inflammatory substances from spinal cord astrocytes and ultimately increase neuronal activity (*Watkins & Maier, 2005*; *Hansson, 2006*). GFAP is the most widely used and a reliable marker for *in vivo* and *in vitro* identification of astrocytes (*Tomassoni et al., 2004*), and represents the major component of intermediate filaments in mature astroglia (*Brenner, 1994*). GFAP has been proven to be important in modulating the motility and shape of astrocytes by giving structural stability to the extensions of cell processes (*Eng, Ghirnikar & Lee, 2000*), in both normal and pathological brain (*Bignami et al., 1972*), and spinal cord tissues (*Song et al., 2016*). Despite its high sensitivity, astrocyte labeling with GFAP antibody is usually accompanied by the detection of another specific marker, called Connexin 43 (Cx-43) (*Ochalski et al., 1996*), which is an important component of astrocyte gap junctions (*Rouach et al., 2002*). Cx-43 maintains the normal shape and function of astrocytes, which is important for their integrity and stability (*Wu, Yu & Feng, 2015*). After peripheral or central nervous damage, Cx-43 expression markedly increases, and its deletion in astrocytes can reduce acute astrogliosis, and can produce analgesia in different pain models (*Gao & Ji, 2010*; *Huang et al., 2012*).

Microglia and astrocyte interaction has been shown to be involved in the establishment and maintenance of pathological pain (*Mika et al., 2013*). Several studies suggest that microglia may be temporarily involved in the initiation of pain, while astrocytic activation may be responsible for its long-lasting maintenance (*Gosselin et al., 2010*). Although it is known that microglia and astrocytes are potential therapeutic targets in pain control, no work has focused on the description of these cells in horses under either normal conditions or during painful states. The use of Iba-1, GFAP and Cx-43 in horses has been documented, mainly in encephalic diseases or in embryotic/fetal development studies (*Siso, Ferrer & Pumarola, 2003*; *Bielefeldt-Ohmann et al., 2017*; *Rigoglio et al., 2017*). However, an improved understanding of glial cell function and morphology in healthy horses is needed to accurately know what happens *in vitro* and *in vivo* in painful syndromes. The objective of this study was to determine healthy spinal microglia and astrocyte morphology and distribution in equine spinal cord dorsal horns using Iba-1 and GFAP/Cx-43 immunofluorescence labeling, respectively.

## MATERIAL AND METHODS

### Animal selection and spinal cord sampling

The Ethics and Bioethic Committee for the Use of Animals in Research of the Universidad Austral de Chile approved this project (No 001/2017). Five gelding mix breeds adult horses

(between four and 12 years of age) were selected from a commercial slaughterhouse (Nueva Imperial Ltda., Temuco, Chile) and were evaluated in the holding pen, immediately prior to humane euthanasia by pneumatic stunning and exsanguination, by an equine clinician. Animals with visible wounds, gait alterations (either musculoskeletal or neurological) or other conditions (pregnant, overweight, foal or senior horses) were excluded. Visual gait analysis was done by an experienced equine clinician. Spinal cord tissues were collected immediately after euthanasia. Lumbar spinal segments (L1–L6) were sampled and sectioned transversely maintaining the integrity of the segments in their entire continuity and then stored in individual jars containing Bouin fixative (75 mL of saturated aqueous picric acid (1.2% w/v), 25 mL of formalin (40% w/v formaldehyde), and 5 mL of glacial acetic acid), according to the method previously described by *Roales-Buján et al. (2012)*. The cranial and caudal aspects of the spinal cord segments were marked using 18G needles.

## Histology and immunofluorescence

Spinal cord segments were fixed for 48 h in Bouin fixative and then separated into 2 mm thick sections. Briefly, sections were first dehydrated with graded ethanol series (70% to 100% for 1 h each, plus an extra hour in 100% concentration), then with a mixture of 100% ethanol and pure butanol (1:1) for 1 h, and finally a dehydration process with pure butanol (two sessions of 1 h each) was performed. These segments were paraffin-embedded for 4 h (divided into four separate sessions, 1 h each at 60 °C), to be later sliced into 6 μm transverse sections using a manual rotatory microtome (Leica Biosystem RM2235; Leica, Wetzler, Germany). Each histological section was dewaxed using pure xylene for 10 min each, rehydrated in alcohol gradient (100% to 70%, 5 min each) and washed with distilled water. For epitope exposure sections were treated with sodium citrate buffer (10 mM Sodium Citrate, pH 6.0) and microwaved three times for 4 min each (700 W). All sections were blocked for 1 h with TCT buffer (0.25% Casein, 0.15 M NaCl, Triton X-100 0.5% in TBS, pH 7.6) at 4 °C. Tissue sections were then rinsed three times for 10 min in Tris-buffered saline (TBS) and incubated with the primary antibody overnight at room temperature (RT). After incubation, tissue sections were washed three times for 10 min in TBS and incubated with the corresponding secondary antibody for 1 h. Three spinal cord sections immediately adjacent were submitted for staining with each biomarker antibody. A total of 15 spinal cord sections for each horse were used for immunofluorescence analysis. For each biomarker, primary antibodies were omitted for negative controls. Hematoxylin and eosin staining was performed in order to identify blood vessels, neurons and check for tissue integrity.

## Single immuno-labeling immunofluorescence

For immunodetection of spinal protoplasmic astrocytes, a polyclonal goat antibody raised against Glial Fibrillary Acidic Protein (GFAP) (1:50; Santa Cruz Biotechnology, Santa Cruz, CA, USA) and a polyclonal rabbit antibody against Connexin 43 (Cx-43) (1:50; Santa Cruz Biotechnology, Santa Cruz, CA, USA) were used. For immunodetection of spinal microglia, a polyclonal rabbit antibody against a synthetic peptide corresponding to the C-terminus of ionized calcium-binding adaptor molecule 1 (Iba-1) (1:50; Santa

Cruz Biotechnology, Santa Cruz, CA, USA) was used. Cx-43 and Iba-1 primary antibodies were incubated with an Alexa Fluor (labeled either with 488 or 594) (1:500; Invitrogen, Camarillo, CA, USA) conjugated goat anti-rabbit secondary antibody for 1 h at RT. Instead, GFAP primary antibody binding was visualized by incubation with Alexa Fluor 488 conjugated donkey anti-goat secondary antibody (1:500; Invitrogen, Camarillo, CA, USA) for 1 h at RT. After immuno-labeling, sections were washed three times for 10 min in TBS and then counterstained with DAPI (1:5000, 4′,6-diamidino-2-phenylindole, dihydrochloride; Invitrogen, Camarillo, CA, USA) for 20 min at RT, washed with TBS and mounted (Fluorescence Mounting Medium; DAKO, Carpinteria, CA, USA).

## Double immuno-labeling immunofluorescence

Co-localization assays were performed to detect spinal cord dorsal horn grey matter astrocytes and microglia (GFAP and Iba-1), and to confirm astrocyte labeling (GFAP and Cx-43). For double labeling with GFAP and Iba-1, tissue slides were initially processed as previously described and after blocking they were processed in the following order: they were first incubated overnight (day 1) with polyclonal goat primary antibody anti-GFAP (1:50; Santa Cruz Biotechnology, Santa Cruz, CA, USA), and were then incubated overnight (day 2) with polyclonal rabbit primary antibody anti-Iba-1 (1:50; Santa Cruz Biotechnology, Santa Cruz, CA, USA). The secondary antibodies were incubated sequentially, first with the conjugated anti-goat secondary antibody (Alexa Fluor 488, 1:500; Invitrogen, Camarillo, CA, USA) for 1 h at RT. After three washes with TBS, the conjugated anti-rabbit secondary antibody (Alexa Fluor 594, 1:500; Invitrogen, Camarillo, CA, USA) was incubated for another hour at RT. Finally, the slides were washed with TBS, counterstained with DAPI (1:5000; Invitrogen, Camarillo, CA, USA) for 20 min at RT, washed again with TBS and coverslip mounted. For double labeling with GFAP and Cx-43, tissue slides were incubated with a polyclonal goat primary antibody anti-GFAP (1:50; Santa Cruz Biotechnology, Santa Cruz, CA, USA) and a polyclonal rabbit primary antibody anti-Cx-43 (1:50; Santa Cruz Biotechnology, Santa Cruz, CA, USA). Finally, secondary antibodies were incubated sequentially as described above (Alexa Fluor 488 conjugated anti-goat secondary antibody and Alexa Fluor 594 conjugated anti-rabbit secondary antibody, and then counterstained with DAPI).

## Image collection and analysis

Iba-1, GFAP and Cx-43 immunoreactivity were analyzed to evaluate spinal astrocytes and microglia morphology and organization. Immuno-labeled sections were visualized at 10X, 20X, 40X, 60X, 63X and 100X magnifications, and examined using an Eclipse E200 Biological epifluorescence microscope and Olympus FV1000 confocal fluorescent microscope. Images were captured with a Basler Scout scA780-541C camera and collected through Pylon Viewer 4 software for epifluorescence microscope images, and laser scanning was performed using Olympus F10 Fluorview software for confocal microscope images. Z-stack of optical sections, 0.5 to 0.9 μm in total thickness, were captured from 15 spinal cord tissue slices from each horse using a confocal microscope. At 10X magnification, an observation of the spinal cord was performed to select those horses with spinal cord sections

that had both dorsal horns completely defined. Finally, five horses met all requirements and were included in the study.

Morphologically, evaluation of shape, area (in spherical form cells) or cell diameter (in ramified form cells), plus process length and thickness was performed using ImageJ. In the case of microglial cell analysis, single cell immune-labeling with Iba-1 positive cells were considered; for astrocytes, double cell immuno-labeling with GFAP (Alexa Fluor 488 labeled) and Cx-43 (Alexa Fluor 594 labeled) was implemented. The diameter of twenty ramified cells was determined measuring the longest axis in non-overlapping cells, with a cross sectional line through the nucleus (DAPI marked) and two endpoint branches. A line between the nucleus and the end of a primary process was traced to measure the process length, and the thickness of these processes was defined with a trace line between two parallel sides of the same process (File S1A). For spherical form cells, cell surface was defined with the free hand selection tool, and a line around twenty cell bodies (DAPI marked) was made to calculate the cross sectional area of glial cells (File S1B).

To determine the number of microglial and astrocyte cells at the dorsal horn, the total number of Iba-1, GFAP and Cx-43 labeled cells detected at 10X magnification in 15 dorsal horn sections were counted. First, due to the large size of the horse spinal cord, several photos ($2,592 \times 1,944$ pixels each) were taken to be later reconstructed using Adobe Illustrator CC 2015 software. Using Image J software, cell counting was performed in a defined square perimeter of 1,000 µm2 (File S1C) in three different segments of the dorsal horn (30 random squares per segment), defined as (1) Posterior dorsal horn (including lamina I, II and III), (2) Intermediate dorsal horn (including lamina IV), and (3) Ventral dorsal horn (including lamina V and IV). All data was collected in Microsoft Excel, and means and standard deviations of each previously defined category were obtained.

# RESULTS

## Morphology and organization of Iba-1 expressing microglia in horse spinal cord dorsal horns

Microglia showed a spherical or amoeboid shape (Figs. 1A–1D), with a mean cell area of $92.4 \pm 34$ µm2. Within the dorsal horn, Iba-1 fluorescence varied in all study subjects. At 10X magnification, the distribution of Iba-1 microglial cells varied among areas within the dorsal horn from the same horse. The higher number of Iba-1 positive cells were found in the posterior dorsal horn, equally reactive in both dorsal horns (Fig. 1E). In all horses, the average number of microglial cells was $9.4 \pm 1.8$ cells for the posterior dorsal horn, $3.8 \pm 3.1$ cells for the intermediate dorsal horn, and $2.5 \pm 2.0$ cells for the ventral dorsal horn. Additionally, a close association between microglia and astrocytes processes was observed using Iba-1 and GFAP double immuno-labeling (Fig. 2B), mainly between microglial cell bodies and secondary astrocyte branches.

## Morphology and organization of GFAP and Cx-43 expressing protoplasmic astrocytes in horse spinal cord dorsal horns

GFAP was mainly found in astrocytic primary processes in comparison to cell somas, highlighting a ramified cell shape (Fig. 2A). Also, most of the evaluated astrocytes had

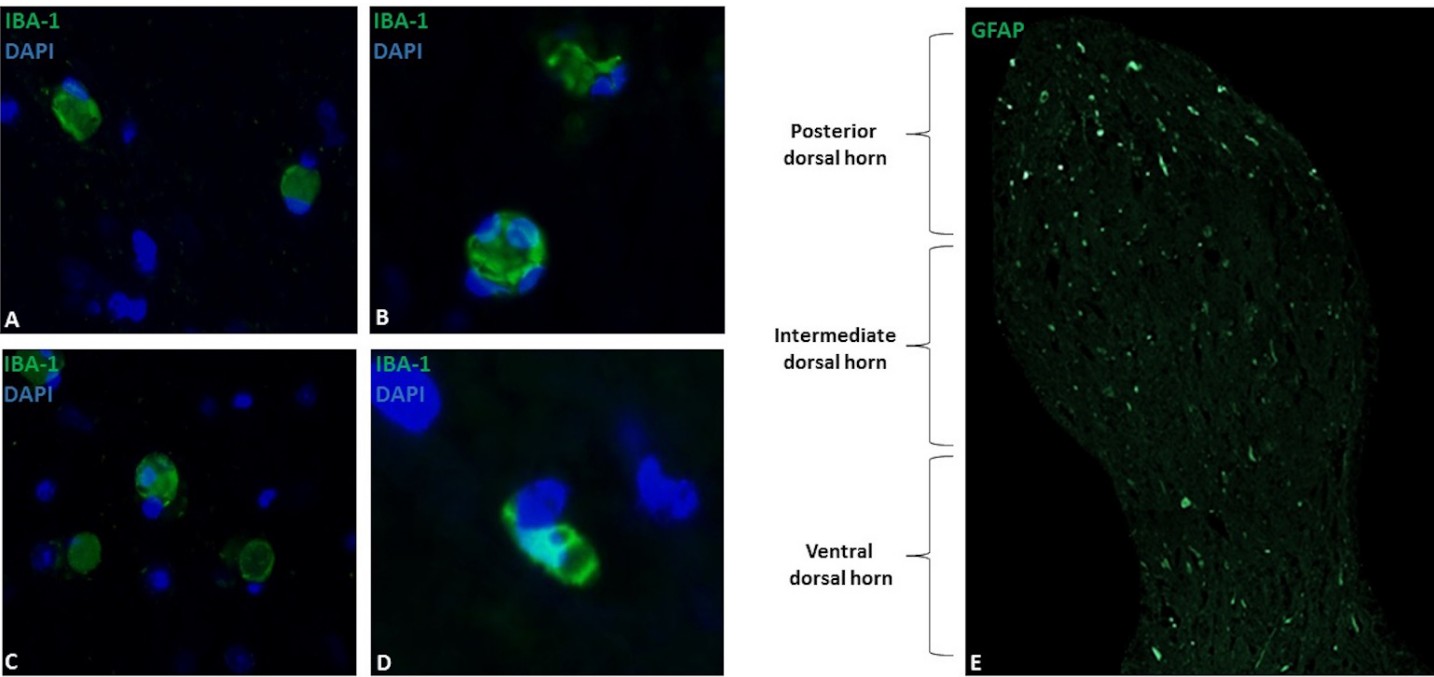

**Figure 1** **Microglial immunostaining against Iba-1 in equine spinal cord dorsal horns (Iba-1, green; DAPI, blue).** Microglia showed a spherical or amoeboid shape, constantly throughout the dorsal horn (63X magnification, epifluorescence microscopy) (A–D). Distribution of Iba-1 marked microglia varied in different areas of the dorsal horn, with a higher microglial cell population in the posterior dorsal horn (i.e., laminae I, II and III) (10X magnification, epifluorescence microscopy) (E). Scale bar, 20 μm ($n = 5$).

two to four primary processes, with a mean of two secondary processes arising from the primary. A close association between secondary processes and neighboring astrocytes and microglia (Fig. 2B), adjacent blood vessels (Fig. 2C) and neurons (File S2) was observed. The arising secondary processes were only visualized using confocal microscopy. Double immunofluorescence with GFAP and Cx-43 was needed to better define astrocyte cell bodies and extensions of processes (Figs. 2D and 2E). According to GFAP and Cx-43 co-localization assays, horse astrocytes have an average diameter of 56 ± 14 μm, with an average length for primary processes of 28 ± 8 μm, and an average thickness of 1.4 ± 0.3 μm. Since it was not possible to visualize the complete cell size at 10X magnification using only GFAP (soma and process could not be defined at this magnification), the cellular density of astrocytes was defined using only Cx-43, which allowed us a better definition of the whole cell in comparison to GFAP. Astrocyte cells were distributed throughout all laminae with a high, moderate and minimal fluorescence. The average number of cells in each segment was: 6.1 ± 2.0 cells for the posterior dorsal horn, 3.6 ± 1.2 cells for the intermediate dorsal horn, and 1.6 ± 1.0 cell for the ventral dorsal horn. Astrocytes were found in cell agglomerations along with other astrocytes, with minimal contact between other groups and a tight connection between end-point processes and capillary walls (Fig. 2F).

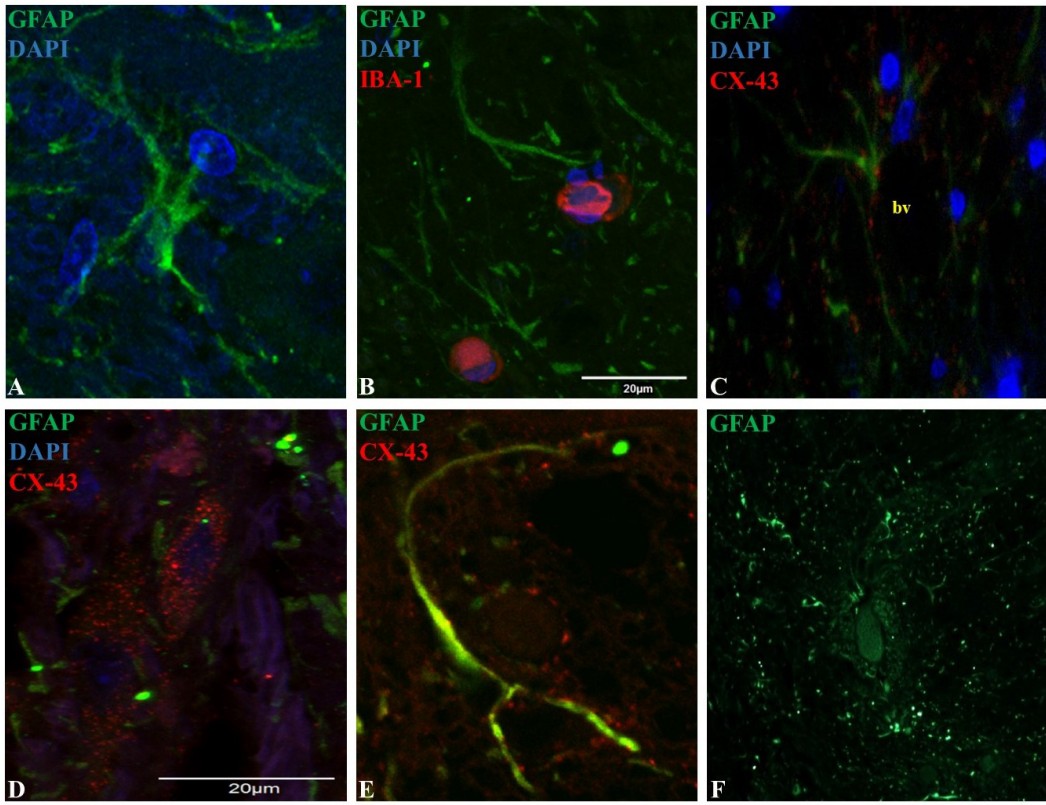

**Figure 2  Representative epifluorescence and confocal images of longitudinal sections of spinal cord dorsal horn in healthy horses.** The sections were double immune-labeled against GFAP (green), Cx-43 (red) and DAPI (blue) to detect astrocytes, and double immune-labeled against GFAP (green), Iba-1 (red) and DAPI (blue) to detect astrocytes and microglial cells. Astrocytes have increased GFAP expression towards the end of their primary processes, and the thin secondary processes arising from the primary processes were in close contact with neighboring astrocytes and microglia (B), vessels (C; bv) and neurons (File S2). Cx-43 was located in cell bodies (D) and processes (E), and astrocytes were found in cell agglomerations along with other astrocytes (F). Images at 60X magnification, scale bar 20 μm (*n* = 5).

## DISCUSSION

It is well known that microglia and astrocytes are potential therapeutic targets for pain control. Therefore, it becomes necessary to understand how these cells maintain and modulate CNS homeostasis (*Old, Clark & Malcangio, 2015*). Several studies have generated a list of glial-derived signaling molecules and mediators involved in the neuron-glia interaction in acute and chronic states of pain (*Ji, Berta & Nedergaard, 2013*). Despite this, there is still a gap in our knowledge about how glial cells are able to alter and maintain states of central sensitization, and even more, mask the analgesic effects of opioids in chronic exposures (*Watkins et al., 2007*; *Old, Clark & Malcangio, 2015*). Until now, the study and detection of these cells has been based on the use of immunomarkers, and of all of them, Iba-1 and GFAP/Cx-43 have been essential to identify microglia and astrocytes, respectively, using immunohistochemical, immunoblotting (Western blot analysis) and

RT-PCR assays, in both clinical and basic neurobiological studies (*Tetzlaff et al., 1988*; *Eng, Ghirnikar & Lee, 2000*; *Tomassoni et al., 2004*).

## Spinal microglial cells in the dorsal horn of healthy horses

In this study, we demonstrated that Iba-1 can successfully immuno-label microglial cells in horse spinal cords, and thus reveal microglia morphology and corresponding distribution within the dorsal horn laminae of healthy horses. Horse microglial cells had an amoeboid and spherical shape (Figs. 1A–1D), mainly located in the same site as in rodents (*Zhang et al., 2008*), in the superficial regions of the dorsal horn (Fig. 1E). Nevertheless, the conventional hyper-ramified shape that is normally visible in resting microglial cells (*Ito et al., 1998*) was not found in any subject. Typically, an amoeboid shape represents active or overactive states of microglial cells after nerve damage (*Eriksson et al., 1993*). Regardless of this, recent studies have shown that microglia's morphological characterization is more flexible and dynamic than previously described, and microglial cell morphology can range from ramified cells to hypertrophied cells with large somas, both in resting and active states (*Zhang et al., 2008*; *Nimmerjahn, Kirchhoff & Helmchen, 2005*). A recent study reported that encephalitic diseases in horses induce an intense branching appearance of microglial cells (*Bielefeldt-Ohmann et al., 2017*). Therefore, there is no evidence that microglial cells in horses have a determined phenotype in both pathological and healthy conditions. This information shows to be valuable when looking for the prototypic phenotype of microglial cells in horses and should be considered. Apparently, the general concept that different states of health can frame a specific cellular form does not exist in horses, just as it has been proven in rodents, suggesting that more than one type of morphology may be found, and thus microglial morphology does not necessarily imply specific functionality. Nevertheless, both studies were performed using a small sample size, and we believe that a specific resting and active microglial morphology cannot be defined.

## Spinal astrocytes in the dorsal horn of healthy horses

GFAP has been extensively used as an astrocyte marker in several mammalian species in addition to humans and rodents (*Machado & Alessi, 1997*; *Nielsen & Jørgensen, 2003*; *Toda et al., 2007*; *Sikasunge et al., 2009*). However, there are no morphological nor structural descriptions of astrocytes in equine spinal cord grey matter.

In our study, we demonstrated that GFAP and Cx-43 can immuno-label spinal astrocytes in horses, and additionally could be used for study of the morphology and distribution of these cells. In this study, a complete description of astrocytes' structure was complex for several reasons. First, GFAP apparently was not detected in the whole cell; and it was mostly expressed in primary processes, and to a much lesser extent in cellular bodies, as it has been described in rodents (*Bushong et al., 2002*; *Blechingberg et al., 2007*). In order to better define astrocyte structure, double immunolabeling of GFAP along with Cx-43 was performed; this permitted a better characterization of the morphology, dimension and cell density (Figs. 2D and 2E). According to our findings, horse protoplasmic astroglia is smaller and structurally less complex showing fewer processes compared to human astrocytes (*Oberheim et al., 2009*). Instead, horse astrocytes show resemblance to those

found in rodents. In terms of cell measurements, we observed that horse astrocytes had an average diameter of $56.0 \pm 14\,\mu m$, with a primary process length of $28.0 \pm 8\,\mu m$, and thickness of $1.4 \pm 0.3\,\mu m$. These same measurements in rodents were $56.0 \pm 2.0\,\mu m$, $37.2 \pm 2.0\,\mu m$, and $2.2 \pm 0.13\,\mu m$, respectively. Therefore, horse and rodent astrocytes seem to have similar cell dimensions, considerably different to those in humans ($142.6 \pm 5.8\,\mu m$, $97.9 \pm 5.2\,\mu m$ and $2.9 \pm 0.18\,\mu m$, respectively (*Oberheim et al., 2009*)). Although horses sit further away phylogenetically than rodents from humans, differences between horses and humans could be affected by type of tissue sample and fixation methods. Therefore, standardization of both factors must be controlled in future studies to determine these morphological differences.

Similarly to microglia, the highest number of astrocytes was counted in the more superficial zone of the dorsal horn (*Ochalski et al., 1996*). In rodents, this segment contains the main termination zone of nociceptive primary afferents (*Todd, 2010*), therefore a correlation between inflammatory/neuropathic conditions and a higher glial cells population could exist and contribute in the maintenance of pain syndromes. Moreover, several studies in humans and rodents explain that spinal astrocytes are organized into domains, where each cell occupies its own anatomical space surrounding a neuronal synaptic space, with process projections that penetrate the process-delimited domains of only a single neighboring astrocyte (*Oberheim et al., 2008*). In this study, we showed some evidence that astrocytes located in the dorsal horn are organized as well defined clusters around blood vessels (Fig. 2F). So far, there is accumulating evidence that the neuron-glia-blood interaction at the spinal cord dorsal horn level is the basis for the inter- and intra-cellular signaling mechanisms that influence synaptic activity, cell growth and nutrition, in addition to the pain signaling that leads to exacerbate chronic and neuropathological pain (*Milligan & Watkins, 2009*; *Sofroniew, 2009*; *Sofroniew & Vinters, 2010*). Nevertheless, more studies are needed to demonstrate the molecular mechanisms involved in the neuron-glia interaction in horses.

## CONCLUSIONS

Although rodents have increased our understanding of pain mechanisms and have helped to predict the effectiveness of certain analgesic mechanisms (*Whiteside, Adedoyin & Leventhal, 2008*), our knowledge of the molecular and cellular mechanisms that underlie chronic pain remains substantially incomplete. Several authors have demonstrated that there are significant differences in the metabolism and anatomy of rodents that can affect extrapolation of data to humans (*Mogil, Davis & Derbyshire, 2010*). Future research should focus on exploring other species that might reveal a glial organization that is closer to human than rodents, in order to investigate the role of glial cells in central sensitization and persistent pain in a more adequate temporal-spatial circumstance. The results described here, including shape, cell dimension and distribution within the dorsal horn of microglia and astrocytes in healthy horses, characterize for the first time the morphology and organizational aspects of horse spinal glia. To our knowledge, this morphologic cell type has not been described before in healthy horse dorsal horns using immunofluorescence

analysis. We believe that a clear understanding of the molecular mechanisms underlying horse glial function is necessary, and could represent a novel approach for efficient novel pharmacological targeting. Upcoming studies will address the molecular and cellular modifications in the neuron-glia network under acute and chronic painful conditions in horses.

## ACKNOWLEDGEMENTS

We gratefully thank Genaro Alvial (Research Assistant at Faculty of Medicine, Universidad Austral de Chile) for his technical support in immunofluorescence analysis.

### Funding

This work was supported by the Dirección de Investigación y Desarrollo, Universidad Austral de Chile, Project S-2014-26. The funders had no role in study design, data collection and analysis, decision to publish, or preparation of the manuscript.

### Grant Disclosures

The following grant information was disclosed by the authors:
Dirección de Investigación y Desarrollo, Universidad Austral de Chile: S-2014-26.

### Competing Interests

The authors declare there are no competing interests.

### Author Contributions

- Constanza Stefania Meneses performed the experiments, analyzed the data, wrote the paper, prepared figures and/or tables, reviewed drafts of the paper.
- Heine Yacob Müller analyzed the data, prepared figures and/or tables, reviewed drafts of the paper.
- Daniel Eduardo Herzberg prepared figures and/or tables, reviewed drafts of the paper.
- Benjamín Uberti performed the experiments, wrote the paper, reviewed drafts of the paper.
- Hedie Almagro Bustamante conceived and designed the experiments, performed the experiments, analyzed the data, wrote the paper, reviewed drafts of the paper.
- Marianne Patricia Werner conceived and designed the experiments, performed the experiments, contributed reagents/materials/analysis tools, wrote the paper, reviewed drafts of the paper.

### Animal Ethics

The following information was supplied relating to ethical approvals (i.e., approving body and any reference numbers):

The Universidad Austral de Chile Ethics and Bioethic Committee for the Use of Animals in Research has approved this project (Nº001/2017).

## Data Availability

The raw data has been provided as a Supplemental File.

## Supplemental Information

Supplemental information for this article can be found online at http://dx.doi.org/10.7717/peerj.3965#supplemental-information.

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
