# Peer review of "Immunofluorescence characterization of spinal cord dorsal horn microglia and astrocytes in horses"

_PeerJ, doi:10.7717/peerj.3965_

## Round 0.1 · original submission · Major Revisions

· Academic Editor

Major Revisions

The peer reviews are positive. Among other isssues , the reviewers deem that a better description of the experimental procedures is needed.Please address all their comments.

Reviewer 1 ·

Basic reporting

This is a reasonably clearly written manuscript.Overall the English is quite good, and only occasional sentences are unclear (as pointed out below). References are appropriate and reasonably up-to-date, and the manuscript overall comply with guidelines regarding structure etc.

Experimental design

In this manuscript, the authors (Meneses et al) describe their studies of distribution, cell sizes and other characteristics of spinal cord dorsal horn microglial cells and astrocytes in horses. The studies are based on material from (only) 5 supposedly healthy, adult horses. However, the selection criteria for the horses are a bit vague and, while that may be due to the “opportunistic” sampling at an abattoir, it is still a concern. Information should, as a minimum, be provided about the estimated age of the horses, their sex and breed. The description of the morphometric assessments (lines 216-226) is somewhat vague and could be greatly aided by some illustrations of the approach (this could be provided in the supplementary file). Moreover, as detailed below, the illustrations of some of the features described cannot be appreciated in the figures provided, notably Figure 2.

Validity of the findings

The authors have a good point about the way the literature regarding shape of microglial cells, as a reflection of activation state, is largely based on dogma originating in some of the early description of these cells, and that the amoeboid shape tends to be the “resting state” in many species rather than the active or reactive state, as originally described. This is certainly the case for species as disparate as mouse, rabbit, ruminants and horses (e.g., J Virol 82: 755; Microb Pathog 103:71). In contrast, the statement that GFAP is not present throughout the astrocyte cytoplasm (line 310-11) is not correct for all species either. It is actually present in such species as mouse and rabbit. What can be concluded from all this is that generalizations cannot be made regarding either of these cell types and that for each species under study appropriate controls must always be applied.
The authors discuss at length the size differences of astroglia between horses and humans (lines 317-25), but do not take into account regions/locations (e.g., the Oberheim paper cited describes neocortical astrocytes). Moreover, differences could be influenced by fixation method used (type of fixative, time of fixation etc) as well as other preparatory methods that could influence tissue shrinkage. This aspect should be commented on.

Comments for the author

The manuscript could be further improved by addressing the following points:
• It is misleading to call the approach “immunohistochemistry”, since a chromogen was not used for visualization of the binding of antibodies. Rather fluorochromes were used and the read-out was immunofluorescence, whether by conventional fluorescent microscopy or confocal. This should be corrected throughout the manuscript, starting with the Abstract (line 31). Furthermore, sections are not “stained” (e.g., line 206), but immune-labeled. Please correct this throughout the manuscript.
• Line 160: what do the authors mean by “mounted in xylene embedded slides”? Please clarify.
• Lines 176 & 178: it is very confusing when the authors say that the primary antibodies were “conjugated” with an [fluorocrome] goat anti-rabbit or donkey anti-goat secondary antibody. The use of the word ‘conjugate’ is misleading in this context, as the term is normally used to describe the joining of two or more chemical compounds, notably the conjugation of an enzyme or a fluorochrome to an antibody. Thus, the secondary antibodies are conjugated with fluorochromes, and the secondary antibodies are used to visualize the binding of the primary antibodies by binding the former to the latter. This should be corrected in the text.
• Figure 2: some of the panels are mis-labeled, i.e., there is no panel D, causing some confusion when reading the results section. In general panels E, F and G (which should be D, E, and F) and in particular panel G (i.e., F) are difficult to interpret. It is not possible to appreciate “a tight connection between end-point processes and capillary walls” (line 268) in the figures provided.
• Lines 41: correct to “amoeboid or spherical”
• Line 58: correct to “found in rodents, with …….”
• Line 108: correct to ‘major’.
• Line 114-115: what do the author mean when saying “that highly coupled these cells”?
• Line 122: correct to “activation at the dorsal horn level”
• Line 128: correct to “under either normal conditions or during painful….”
• Line 130: correct author name to “Bielefeldt-Ohmann” as in the reference list.
• Line 134: replace “disposition” with ‘distribution’.
• Line 141: delete “has”.
• Line 281: delete the second “in”.
• Line 284: replace “stain” with ‘label’
• Line 294: correct to “resting and active states”
• Line 297: replace “o” with ‘and’.
• Line 308: correct “immunostain” to “immunolabel”
END

Reviewer 2 ·

Basic reporting

No comment.

Experimental design

No comment.

Validity of the findings

No comment.

Comments for the author

The authors described the morphological studies of microglia and astrocytes in horse spinal dorsal horn with immunohistochemistry in this manuscript titled 'Immunohistochemical characterization of spinal dorsal horn microglia and astrocytes in horses'. Since the whole study was based on Iba1 and GFAP antibodies, therefore I would recommend authors to use the best well-known antibodies for Iba1 (WAKO) and GFAP (DAKO) instead of the antibodies from Santa Cruz. And the staining for glia cells in the manuscript was not good. As we know that the fixation could also affect the glia cell staining, where Iba1 and GFAP antibodies show perfect staining with mild fixed tissues. And the authors used around 10% PFA for immersion fixation tor two days, which would definitely reduce the staining quality (usually 4% PFA for 2 hrs of immersion fixation).

Reviewer 3 ·

Basic reporting

English is fine and there is adequate background literature presented to justify studying the topic, but that material leads the reader to think a somewhat different study will be done. A bit more background is needed for the specific work that was actually done and a bit more justification is needed for what actually was done. The general structure of the article is fine. There is no concrete hypothesis stated. See General Comments section below for suggested improvements.

Experimental design

The research appears to be original within the Aims and Scope of the journal. However, the research question needs to be more well defined as does the research gap being filled. Portions of the Methods need to be described in more sufficient detail for replication. See General Comments section below for suggested improvements.

Validity of the findings

Clearer description of some of the Methods would make the data more robust and validity easier to evaluate. The conclusions could be more tightly linked to the major issue presented, which is the substrates of pathological pain states. See General Comments section below for suggested improvements.

Comments for the author

INTRODUCTION

A significant issue with the Introduction is that it leads the reader to believe that the authors will be doing a study that examines the interaction of microglia and astrocytes underlying pain syndromes when in reality they are simply looking at the normal morphology of microglia and astrocytes in horse. In essence, the last paragraph could serve as the principal portion of the Introduction if the authors just talked a bit about the glial markers in that particular paragraph, talked a bit more about the existing work in the horse, and provided a more detailed explanation of how this normal data will be useful in future pain studies.

Some more specific comments regarding the Introduction:
In line 68, the concept of the term “static” is a bit vague.

For the sentences that begin on lines 73 and 74, in-text examples would be useful (aside from simple referencing).

Perhaps “differential functionality” might be better at the beginning of line 77.

Line 78: maybe “duration” instead of “time”.

Line 79: “Glial-derived” might be better.

In the sentence beginning on line 83, it’s not clear whether the authors are suggesting pain syndromes cause the microglial response or whether the response underlies the syndrome. So maybe the beginning should read “Painful syndromes involve the triggering of”. Between this sentence and the following one (beginning on line 86), there is unnecessary repetition regarding Iba-1.

Line 89: This sentence needs some clarification.

Line 100: what do the authors mean by housekeeping function? This sentence seems to have some circular reasoning (e.g. resting astrocytes express basal levels of GFAP, but how do you know they’re basal levels unless you first know they’re in the resting state? By GFAP levels? See what I mean?).

Lines 101/102: Please clarify “external stimulation”.

Line 103: What do the authors mean by “pre-synaptic space”?

Lines 102-106: The sequence of interaction between neuron and astrocyte could be clarified here.

Line 106: Perhaps change “at the spinal level” to “at the spinal cord”.

Line 108: “major” instead of “mayor”.

Line 112: Might add “with GFAP antibody” after “astrocyte”.

Line 120-122: This sentence needs to be clarified.

Lines 125-126: This sentence sounds like the goal of this study, given the preceding Introduction material.

MATERIALS AND METHODS

The description of the sectioning and staining are fairly straightforward. However, the description of the image collection and analysis is much less clear/detailed and can use significant improvement, particularly how the samples from a given horse were assigned to different staining and analysis procedures, and the description of the various morphological measures. More specific comments follow:

For animal selection, indicators of pain, which would be important in such a study, were of the most general nature.

Why were either cervical or lumbar samples used from a given horse and not both?

There are no references for the tissue fixation and staining. How did the authors derive those procedures?

How were the sections parsed for the different staining procedures?

The logic for the double antibody staining should be explained more clearly.

Lines 154/155: Transverse sections? 70%, 80% 90%, etc. or 70%, 75%, 80%, 85%, etc.?

Line 159: Do the authors really mean sagittal? If so, why?

Line 160: Model of Jung microtome? What is meant by xylene embedded slides?

Line 162: Please see query for lines 154/155.

Line 163: What type of microwave and what were the settings?

Lines 164/165: Was there some kind of serum used along with the TCT buffer for blocking?

Line 176 and 178: Do authors mean “incubated” instead of “conjugated”.

Line 179: Was there any wash after the secondary antibody incubation before DAPI staining?

Line 180: What form of DAPI was used? Aren’t there a few different types?

Line 188: Regarding “initially processed as previously described”, up to what point?

Line 189: It’s not clear what “incubated separately” means regarding the primary antibody double-staining.

Line 203: Should probably add, “and then counterstained with DAPI”.

Line 210: Might add “was performed” after “laser scanning”.

Line 212: Are the 15 sections all the horses combined, or 15 sections from each horse?

Line 213: It’s not clear what the authors mean by “the entire spinal cord”.

Line 214: Please clarify what is meant by “completely integrated dorsal horns”.

Lines 216-226: The description of the morphological parameters that were measured should be explained more clearly and with more detail. For example, how was shape determined? How was the decision made to classify a cell as spherical in form or hyper-ramified? Was the length of all processes of a cell measured? Did diameter of hyper-ramified cells include the processes? A figure would be helpful showing the various morphometric measures that are described. When measuring process length, was bending of the process taken into account at all?

Line 224: Should the word “cells” be “processes”?

Line 227-234: Please elaborate on what is meant by “distribution” in line 227. How was the dorsal horn and its laminae identified in these fluorescence sections? What was the criterion for an immunopositive cell at 10X magnification? Was it any stained piece that was on the perimeter of the defined square? How thick was the perimeter line?

There is no mention of background staining in the omission controls. How was background staining dealt with?

RESULTS

It is difficult to assess the validity of the data without clarifying some of the issues raised above for the Materials and Methods, particularly with respect to how tissue slices were assigned to the various staining procedures for each animal.

A table summarizing the results would be useful.

Is there a way to know whether the microglia are all amoeboid/spherical, or whether processes just can’t be detected by the staining method?

When talking about differences between any measures (e.g. between numbers of cells in dorsal horn regions), the authors should note they are simply numerical differences as opposed to statistically significant differences, since no statistical comparisons were made.

Is there a reason why the microglial/astrocyte branch contacts weren’t quantified? Perhaps they differ between dorsal horn regions.

How were primary and secondary branches defined?

Many of the descriptions are quite qualitative and it’s hard to evaluate how representative they really are.

Lines 252-254: The authors should clarify what they mean when they say, “GFAP was mainly found in astrocytic primary processes”. Compared to what? Small cell bodies relative to what? How were “long branches” defined when the authors say “extended 2-4 long branches”?

Lines 254-257: So primary astrocyte processes then give off 2-4 long branches (why weren’t descriptive statistics calculated here?), which each give off 2 small thin branches? Correct? How were the blood vessels and neurons that the authors mention identified?

Lines 258-259: How did double fluorescence make it better for defining astrocyte cell bodies and extensions of processes?

Lines 259-261: The authors should explain why it was necessary to use diameter measurements for cell body size in astrocytes but area for cell body size in microglia. How was “main process” defined?

Lines 261-263: The authors should clarify why they needed to see the whole cell size for the astrocyte density measurement, and how the Cx-43 staining help with that.

DISCUSSION

The Discussion makes some interesting points regarding differences with other species and with humans. However, the authors do not really explain how these particular findings are relevant to deciphering the substrates of pathological pain states. Also, no limitations of the study are discussed and there is no mention of effects of background staining.

Lines 286-287: Microglial cells were also found in the other dorsal horn regions examined. They were just numerically less dense.

Lines 288-289: Maybe no processes were observed using Iba-1 because of staining quality.

Lines 299-302: The data you talk about in this paragraph actually do seem to support a difference in microglial morphology between healthy and (encephalitic) diseased horses.

Why did the authors not discuss the findings on dorsal horn laminar distribution of microglia?

Lines 317-325: The horse, rat human comparisons are interesting, but what does it mean for deciphering mechanisms of pathological pain states? Same question for the laminar distribution?

Line 326: This statement should be toned down given that no comparative statistics were performed.

Lines 330-333: It is hard to see what the authors describe here in Figure 2F, and the statement about “stretch connection” is not clear.

Lines 333-336: The authors need to elaborate a bit here in order to clarify the statement.

CONCLUSIONS

Lines 344-348: Perhaps the authors might say that exploring other species might reveal glial organization that is closer to human than is rodent, and might represent better models for investigating glial substrates of pathological pain.

Lines 353-355: So are you saying you feel horse is a better model for human pathological pain states than rodent?

---

## Round 0.2 · Minor Revisions

· Academic Editor

Minor Revisions

The manuscript has been much improved. Please correct the remaining language/grammatical issues and provide the clarifications requested by reviewer #3

Reviewer 1 ·

Basic reporting

Only minor spelling and syntax issues still need to be corrected. Otherwise much imporved.

Experimental design

No further comments

Validity of the findings

No further comments

Comments for the author

The Authors should be commended for their attention to the reviewers' comments and criticisms.The manuscript is now notably improved, and only minor issues remain to be dealt with:

line 53: correct to 'immuno-label'
line 202: correct to 'three'
line 216: suggest modify the sentence to the following "GFAP primary antibody binding was visualized by incubation with Alexa Fluor 488 conjugated donkey anti-goat secondary antibody (1:500, Invitrogen, Camarillo, CA, USA)...."

line 339: delete "to obtain"
line 394: replace "o" with "and"
line 397: change "evidencing" to "suggesting"
line 406: correct to "used for study of the morphology......"
line 507: correct to "focus on exploring...."
line 530: change to "immunofluorescence analysis".

END

Reviewer 3 ·

Basic reporting

Please note that this is a re-review of the original submission. The basic reporting is now satisfactory and no longer misleads the reader to think that a different type of study will be done.

Experimental design

The descriptive nature of the study has now been made clearer and many of the methodological details lacking in the original have been added or clarified. However, there are still a few details that could use clarification that are noted below.

Validity of the findings

As noted, several methodological details have been clarified, improving the perceived validity of the findings. The authors have added a few theoretical connections between their findings and pathological pain states, but the discussion of limitations of the study is still minimal.

Comments for the author

The authors have addressed the vast majority of the comments made by this reviewer on the original submission.

INTRODUCTION:
The Introduction now seems adequate.

MATERIALS AND METHODS:
The reference that the authors cite in their responses, for the Bouin’s fixation procedure, should be added near the end of paragraph 1 of the Methods.

The author’s response describing the 3 adjacent sections for the 3 antibodies and the total of 15 sections per horse should be added to the Methods text.

In the Methods section it still needs to be made clearer why there seems to be a set of single labeled tissue produced, and then another set of double labelled tissue produced, despite the fact that the authors feel it’s clear.

Line 167: The authors should further clarify what “separately” means (is it really sequentially, with first primary antibody on Day 1 and second primary antibody on Day 2?).

Line 198: How was a hyper-ramified cell defined?

Line 207: When the authors say GFAP and Cx-43, do they mean double labelled, while the Iba-1 is single labelled tissue? Irrespective, this should be clarified in text.

Lines 212-215: The authors say in their response letter that dorsal horn laminae were not identified, so how did they make the laminar distinctions mentioned in these lines and how were they able to visualize dorsal horn regions?

It’s still not clear how the authors can have confidence in their numbers without a background estimate/correction.

For Supplemental File A: the numbers in Part A should be explained. Line 12 should read “using 30 square perimeters”.

RESULTS
For the qualitative descriptions in lines 225-227, and 234-235, some idea of how representative this is should be presented. Otherwise, it’s hard to interpret the validity of the observations.

Line 232: Why is the number of primary processes presented as a range and not a mean?

Line 242: Cx-43 alone?

The HE staining comment in the author’s response regarding tissue integrity and blood vessels should be added to the Methods.

DISCUSSION
The author’s response regarding lines 299-302 of the original submission should be placed in the Discussion around line 275.

Lines 278-279 need to be toned down because the very limited data available in horse, although inconclusive due to small numbers, seem to indicate there might be a relation between disease state and cell form.

Line 306: Should say, “…the highest number of astrocytes were counted in laminae I, II and III…”

Line 313: Should say, “…we showed some evidence that astrocytes…”

---

## Round 0.3 · accepted · Accept

· Academic Editor

Accept

I am satisfied with your changes and am glad to approve your manuscript for publication.

The decision of whether or not to publish the peer reviews alongside the paper is entirely yours, and will not affect how your paper is handled going forward. However, I encourage you to do so. Making the reviews public allows the reviewers to receive more credit for their efforts, and also contributes to the emerging culture of fairness and transparency in editing and peer review.